# Identification of Therapeutic Targets in an Emerging Gastrointestinal Pathogen *Campylobacter ureolyticus* and Possible Intervention through Natural Products

**DOI:** 10.3390/antibiotics11050680

**Published:** 2022-05-18

**Authors:** Kanwal Khan, Zarrin Basharat, Khurshid Jalal, Mutaib M. Mashraqi, Ahmad Alzamami, Saleh Alshamrani, Reaz Uddin

**Affiliations:** 1PCMD, International Center for Chemical and Biological Sciences, University of Karachi, Karachi 75270, Pakistan; khankanwal011@gmail.com (K.K.); riaasuddin@yahoo.com (R.U.); 2Jamil-ur-Rahman Center for Genome Research, Dr. Panjwani Center for Molecular Medicine and Drug Research, International Center for Chemical and Biological Sciences, University of Karachi, Karachi 75270, Pakistan; zarrin.iiui@gmail.com; 3HEJ Research Institute of Chemistry, International Center for Chemical and Biological Sciences, University of Karachi, Karachi 75270, Pakistan; 4Department of Clinical Laboratory Sciences, College of Applied Medical Sciences, Najran University, Najran 61441, Saudi Arabia; mmmashraqi@nu.edu.sa (M.M.M.); saalshamrani@nu.edu.sa (S.A.); 5Clinical Laboratory Science Department, College of Applied Medical Science, Shaqra University, Al-Quwayiyah 11961, Saudi Arabia; aalzamami@su.edu.sa

**Keywords:** pan-genome, *Campylobacter ureolyticus*, UDP-3-O-acyl-N-acetylglucosamine deacetylase, LpxC, campylobacteriosis

## Abstract

*Campylobacter ureolyticus* is a Gram-negative, anaerobic, non-spore-forming bacteria that causes gastrointestinal infections. Being the most prevalent cause of bacterial enteritis globally, infection by this bacterium is linked with significant morbidity and mortality in children and immunocompromised patients. No information on pan-therapeutic drug targets for this species is available yet. In the current study, a pan-genome analysis was performed on 13 strains of *C. ureolyticus* to prioritize potent drug targets from the identified core genome. In total, 26 druggable proteins were identified using subtractive genomics. To the best of the authors’ knowledge, this is the first report on the mining of drug targets in *C. ureolyticus*. UDP-3-O-acyl-N-acetylglucosamine deacetylase (LpxC) was selected as a promiscuous pharmacological target for virtual screening of two bacterial-derived natural product libraries, i.e., postbiotics (*n* = 78) and streptomycin (*n* = 737) compounds. LpxC inhibitors from the ZINC database (*n* = 142 compounds) were also studied with reference to LpxC of *C. ureolyticus*. The top three docked compounds from each library (including ZINC26844580, ZINC13474902, ZINC13474878, Notoginsenoside St-4, Asiaticoside F, Paraherquamide E, Phytoene, Lycopene, and Sparsomycin) were selected based on their binding energies and validated using molecular dynamics simulations. To help identify potential risks associated with the selected compounds, ADMET profiling was also performed and most of the compounds were considered safe. Our findings may serve as baseline information for laboratory studies leading to the discovery of drugs for use against *C. ureolyticus* infections.

## 1. Introduction

*Campylobacter ureolyticus* is a Gram-negative bacterium previously classified as *Bacteroides ureolyticus* and belongs to a class of pathogens that cause gastroenteritis [1]. *Campylobacter* outnumbers *Shigella*, *E. coli O157*, and *Salmonella* as the most common cause of bacterial enteritis [2]. Previously, the most common *Campylobacter* species linked with human illness were *C. jejuni* and *C. coli*, but breakthroughs in molecular diagnostics paired with the development of innovative culture methods have helped identify and isolate a range of under-reported and fastidious *Campylobacter* species, including *C. ureolyticus*. Farm animals are the primary reservoir for *Campylobacter* sp. infections and the primary cause of campylobacteriosis. Farm animals are also the leading source of bacterial food poisoning and *Campylobacter* gastrointestinal illnesses worldwide. Campylobacter foodborne illness is a concern and an expensive burden for the human population, accounting for 8.4% of all diarrhea cases worldwide. In most cases, *C. ureolyticus* has been recovered from human samples, with just one report of *C. ureolyticus* isolation from healthy horse endometria. Following that, a retrospective investigation of over 7000 patients with diarrhea found *C. ureolyticus* in 23.8% of *Campylobacter*-positive samples, marking the first discovery of *C. ureolyticus* in the faeces of gastroenteritis patients and highlighting the species’ status as an emerging enteric pathogen [3,4,5].

*C. ureolyticus* has been accompanied by a variety of illnesses in the past, including bacterial vaginosis [6], gangrenous lesions [7], superficial ulcers [1,8], nongonococcal urethritis [1,9], male infertility [10] and more recently, ulcerative colitis [11]. Furthermore, *C. ureolyticus* has been associated with periodontal diseases, such as gingivitis and periodontitis, similar to numerous other new and atypical *Campylobacter* species [9]. Recent research has led to the discovery and isolation of *C. ureolyticus* as the sole pathogen in the feces of many diarrheal patients. It is now thought to be the second most common *Campylobacter* species found in patients suffering from diarrhea, surpassing the established pathogens *C. coli* and *C. jejuni*. Furthermore, an examination of infectivity data indicates the predominance of this pathogen in diarrheal patients between the ages of 5 and 70, suggesting that it impacts the pediatric population more than adults. Patient data for gastrointestinal illnesses suggest that *C. ureolyticus* is an emerging gastrointestinal pathogen [11]. 

Its pathogenesis is based on invasion, colonization, adhesion, and toxin release. Previously, the absence of genetic data hindered the understanding of in-depth pathogenic mechanisms and virulence, but the democratization of next-generation sequencing (NGS) has made it easier to explore the species at genome scale [12]. Pan-genomic studies help exploit the entire repertoire of genes from all strains within a species, which enables investigation of genomic diversity and similarity [13]. This type of information aids the understanding of species evolution and pathology and helps identify drug targets for pan-treatment of *C. ureolyticus* [7,14]. In light of this, the current study aimed to investigate multiple isolates of *C. ureolyticus* to better understand its pathogenesis and characterize the core and pan-genome subsets to prioritize novel therapeutic targets from shared core genes. To the best of our knowledge, this is the first pan-genome study on *C. ureolyticus*, as well as the first study to make use of core gene data to exploit therapeutic targets. Virtual screening was also carried out against a selected target, using natural product libraries to identify potent inhibitors that could prove useful for campylobacteriosis treatment. 

## 2. Results

### 2.1. Pan-Genome and Resistome Analysis 

The pan-genome of 13 *C. ureolyticus* strains was composed of ~5500 accessory, 1182 core, and 400 unique genes. The pan-genome curve showed a B_pan_ value of 0.17 (i.e., <1), representing the open nature of the pan-genome of this pathogen (Figure 1A). The comparative study showed that each strain shared ~300–500 genes in the accessory genome fraction, with the highest number retrieved for the strain NCTC10941 (*n* = 535 genes), while strain ACS-301-V-Sch3b harbored the maximum number of unique genes (*n* = 114 genes). Additionally, five strains (ACS-301-V-Sch3b, LFYP111, LMG 6451, RIGS 9880, and UMB0112) lacked certain genes that were present in other strains (Appendix A). The COG functional analysis provided insights into the conserved proteins and their specific metabolic pathways. The core genome showed the highest number of genes pertaining to metabolism, while the accessory genome contained the largest fraction of genes related to cellular processes and signaling. Further analysis revealed that the core genome was enriched in amino acid transport/metabolism, translational, ribosomal structure/biogenesis, and energy production/conversion apparatus, while the accessory genome was enriched in the cell wall, membrane, envelope biogenesis and inorganic ion transport metabolism. The unique genome comprised the main fraction of genes related to information processing and storage and those of unknown function (Figure 1B). 

Additionally, the phylogenetic tree generated for the pan- and core genomes highlighted monophyly in all 13 strains (Figure 2). Strains with greater proportions of shared genes have been found in close lineages, hence the proportion of pan-genes between them reveals their evolutionary closeness. In *C. ureolyticus*, the average proportion of common genes was ~61.86 percent. Phylogenetically, all strains represented in the pan- and core genomes were found to be a group in almost the same clade, indicating similarities.

The resistome analysis of these 13 strains resulted in the identification of certain antibiotic resistance genes (ARGs), found in the core and accessory genomes but absent in the unique genome fraction. The core genome had only the *gyrA* resistance gene, which encodes for fluoroquinolone via antibiotics that target the alteration mechanism [15]. Additionally, it was observed that the *gyrA* gene contained a single-nucleotide polymorphism at position T86I. This mutation has previously been well characterized in *Campylobacter* species [16,17,18,19] and has been tested as a biomarker to detect fluoroquinolone resistance in *C. jejuni* [20]. However, no such report for *C. ureolyticus* exists yet. In the accessory genome, two resistance genes, OXA-85 and TetM, were identified, which encode for OXA-beta-lactamases and tetracycline-resistant ribosomal protection protein. These proteins showed resistance to carbapenem, cephalosporin, penam class, and tetracycline antibiotics through antibiotic inactivation. OXA-85, encoded by FUS-1, is a narrow-spectrum beta-lactamase. It has been isolated previously from *Fusobacterium nucleatum* subsp. *polymorphum* [21] but not yet reported in any *Campylobacter* spp. This means that this gene could have been introduced in *C. ureolyticus* via horizontal gene transfer. TetM has been widely reported in many species before [22,23,24,25,26,27] and may be transposon- or chromosomal-mediated [25,27]. 

### 2.2. Differential Sequence Mining and Therapeutic Target Identification

The identified 1882 core genes of *C. ureolyticus* were further used for downstream drug target identification. To prioritize potent drug targets, paralogous, essential, and non-homologous sequence identifications are the crucial steps. CD-Hit redundancy analysis was used to eliminate paralogous sequences from the identified core fraction data, resulting in the discovery of 1178 paralogous sequences. The role of essential proteins helps in the survival of the pathogen. We used BLASTp with an E-value of 10^−5^ to locate the translated gene product against the DEG and CEG databases. This resulted in the shortlisting of 757 and 659 essential proteins, respectively. The intersection of this dataset revealed 647 common proteins. Furthermore, there were 109 *C. ureolyticus* proteins non-homologous to the gut bacterial proteome. These proteins were involved in critical cellular and metabolic processes. Selective targeting of these proteins could prevent cytotoxic reactions and harmful effects in the human host. 

The ability of a small-molecule medication to regulate the activity of a therapeutic target is known as druggability. Proteins with a high frequency of sequence similarities (80% or more) were considered as druggable targets. Among 109 proteins, 35 were identified as druggable. These targets were then examined for their virulence as well. Among these, 26 proteins were found to be virulent (Table 1) and belonged to either metabolism, information or signaling, and cellular process families of proteins. 

UDP-3-O-Acyl-N-acetylglucosamine deacetylase (LpxC) was selected as a therapeutic target. It is a zinc-dependent metalloamidase and catalyzes the second and final step of lipid A production [28]. It eradicates the acetyl group from UDP-(3-O-(R-3-hydroxymyristoyl))-N-acetylglucosamine and results in the production of UDP-(3-O-(R-3-hydroxymyristoyl))-glucosamine and acetate. The ensuing enzymatic reaction converts UDP-(3-O-(R-3-hydroxymyristoyl))-glucosamine to lipid A, which is later integrated into lipopolysaccharides [29]. It is an appealing and validated target for the development of novel antibacterial medicines to treat Gram-negative infections because of its crucial role in lipid A production. Due to the absolute dependency of the microbes on this biosynthetic pathway, along with its nonexistence in humans [30], this potential drug target was chosen for further processing in *C. ureolyticus*. 

### 2.3. Structure Prediction and Inhibitor Screening 

The structure of *E. coli* LpxC (PDB ID: 4MDT) was utilized as a template due to its high sequence identity of 44 percent with the LpxC of *C. ureolyticus*. A 3D stereochemical model (Figure 3A) revealed 95% residues in the core favorable area (Figure 3B) and 2.1% outliers, with a Z-score of 0.78. Around 26% of the proteins consisted of alpha helices and 38% of beta-sheets, while 8% were disordered. Since the metal cofactor is important for LpxC catalytic activity, the zinc ion was also kept bound with the structure for docking.

Many effective LpxC inhibitors have been identified [31,32], with a range of chemical scaffolds and antibiotic profiles [31], but none of them has attained approval as an antibacterial moiety [31]. LpxC inhibitors have not been reported for *C. ureolyticus*. For this purpose, therefore, structure-based inhibitor screening was attempted using a molecular docking strategy. This is an excellent method for determining how drugs/compounds interact with a biological target. To comprehend the LpxC-drug binding mechanism and energy, analyses were conducted in MOE software. Compound libraries included LpxC inhibitors from the ZINC database, metabolites from *Streptomyces* spp., and postbiotics. *Streptomyces* spp. produce the largest number of antibiotics among all bacteria [33]. They decrease the fitness costs of secreted secondary metabolites, leading to enhanced yield and product diversity [34]. Postbiotics are the healthful metabolites of probiotics [35]. The use of such metabolites has been reported against *Vibrio campbelli* infection [36] and SARS-CoV-2 [37]. Docking revealed the binding interactions for the top selected compounds from each library (Table 2). 

ZINC26844580 (S-value: −7.42), ZINC13474902 (S-value: −7.05), and ZINC13474878 (S-value: −6.90) were prioritized from the ZINC database LpxC inhibitors. Notoginsenoside St-4 (from *Lactobacillus caseii*, S-value: −8.59), Asiaticoside F (from *Lactobacillus caseii*, S-value: −8.43), and Paraherquamide E (from *Lactobacillus plantarum*, S-value: −8.02) were shortlisted from the postbiotics library, whereas ZINC08219868 (Phytoene, S-value: −7.20), ZINC08214943 (Lycopene, S-value: −7.03), and ZINC04742519 (Sparsomycin, S-value: −7.01) were selected from the streptomycin library. Figure 4 shows the 2D bonding interactions between the shortlisted compounds and the LpxC protein. Among the prioritized compounds, Notoginsenoside St-4, a dammarane-type saponin from the root of *Panax notoginseng* has been previously reported to hamper herpes simplex virus entry [38]. 

Asiaticoside F has also been reported from the leaves of *Centella asiatica* and is known to inhibit tumor necrosis factor-α [39]. Paraherquamide E has been isolated from the fungus *Penicillium charlesh* and is a known anti-nematicidal agent [40]. These postbiotics have previously been isolated from plants or fungi, but bacterial production of this compound heralds an exciting use of biogenic metabolites of probiotic strains. 

Phytoene is produced by *Streptomyces scabrisporus* NF3 [41], *Streptomyces griseus* [42], archaeon *Thermococcus kodakarensis* [43], algae *Dunaliella* sp., [44], yeast *Xanthophyllomyces dendrorhous* [45] and halophilic bacteria [46], among other microorganisms. Lycopene has also been produced by microorganisms [47]. Phytoene and lycopene have radical scavenging activity [48]. Sparsomycin has also been previously isolated from *Streptomyces* spp. and possesses anti-tumor activity [49]. Inhibition of LpxC by these compounds is of interest, making for welcome additions to the class of existing inhibitors of this enzyme.

### 2.4. ADMET Profiling of Shortlisted Compounds

The majority of the shortlisted compounds were not found to inhibit the P450 class of enzymes, CYP1A2, CYP2C19, CYP2C9, CYP2D6, and CYP3A4. ZINC13474902 and ZINC13474878 inhibited several cytochrome P450s. Since these enzymes are involved in drug metabolism [50], deactivating or excreting them, their non-binding behavior indicates that the shortlisted drugs will not be rendered ineffective. Six compounds (ZINC26844580, ZINC13474902, ZINC13474878, Notoginsenoside St-4, Asiaticoside F, and Sparsomycin) were identified as non-permeable to the blood–brain barrier, while three (paraherquamide E, phytoene, and lycopene) were permeable. GI absorption was low for Notoginsenoside St-4, Asiaticoside F, and Sparsomycin. Skin sensitization was not observed, while significant permeability to Caco2 was seen. Caco2 being used as a model to study the intestinal displacement barrier, this means that the metabolites can cross the intestinal barrier. Two compounds (Notoginsenoside St-4 and Asiaticoside) were identified as P-glycoprotein inhibitors. This could have an impact on the tissue distribution of these compounds and may change the pharmacokinetics due to the drug–drug interaction properties of P-glycoprotein inhibitors [51]. Except for Notoginsenoside St-4 and Asiaticoside, all compounds followed Lipinski’s rule of five. Compounds were non-toxic and AMES carcinogenicity was null (Table 3), except for ZINC13474878. These features make several of the screened inhibitors ideal candidates for future therapeutic development. For the compounds that do not fall in the ideal category of drug-likeliness or safety profile, their scaffolds could be used to make better compounds. Additionally, a conjugate could be made for the compounds that have low gastrointestinal tract absorption.

### 2.5. MD Simulation Analysis

A classic MD analysis was performed to determine the free energies of complexes, i.e., MM/PBSAs were almost similar, with the lowest value for the LpxC–lycopene complex (Table 4). The atom-scale MD and the structural stabilities of the selected inhibitors and protein complex were determined in a time-dependent manner. RMSD, RMSF, hydrogen bonding, and Rg of the ligand-bound LpxC protein were studied to analyze the structural integrity during bonding. This was achieved through trajectory mapping graphs of the backbone carbon atoms. 

The RMSD analysis for ZINC LpxC database compounds (ZINC26844580, ZINC13474902, and ZINC13474878) showed the stability of compounds during the whole 50 ns simulation study, with a mild fluctuation found in ZINC26844580 at ~30 ns. The complete system was found to be at equilibrium within the range of 0.15–0.20 nm, as is evident from Figure 5A. The RMSD analysis for the postbiotic shortlisted compounds (Asiaticoside F and Paraherquamide E) showed considerable fluctuation during the initial 10 ns of simulation. However, after 15 ns, the system progressively stabilized and remained stable until the simulation was completed, with an average deviation of 0.15–0.20 nm, indicating the simulated system’s convergence. Notoginsenoside St-4 showed stability throughout the simulation study, as is apparent from Figure 5B. Compounds from the streptomycin library (ZINC08219868 (Phytoene), ZINC08214943 (Lycopene), and ZINC04742519 (Sparsomycin)) exhibited mild to moderate fluctuations during the 50 ns simulation, showing variations in RMSDs values in an acceptable range of 0.15–0.25 nm (Figure 5C). 

However, to evaluate the fluctuations found in amino acid residues of the ligand-bound protein during the simulation studies, RMSF data for these compounds were also plotted. Figure 6 depicts the RMSF plot for all nine shortlisted compounds, which displayed variations throughout the simulation. The average RMSF for the simulated system was determined to be 0.6 nm. RMSF was considerably stable for all of the amino acid residues of the ligand-bound protein in the simulated system. 

To evaluate the compactness of the structure complexes, Rg was plotted (Figure 7). The Rg plot for the catalytic site during the initial 10 ns of simulation showed large fluctuations. However, after 20 ns, the system gradually moved towards compaction, contributing to the overall stability of the simulated protein structure in the presence of all shortlisted compounds. 

The interactions between the protein and compounds were further determined through hydrogen bonding analysis. A robust interaction was observed between ZINC26844580, ZINC13474902, ZINC13474878 (ZINC LpxC inhibitor compounds), Asiaticoside F, Paraherquamide E, Notoginsenoside St-4 (Postbiotics), and LpxC protein in terms of hydrogen interaction, ranging from 8-10 bonds throughout the 50 ns simulation. The compound ZINC04742519 (Sparsomycin) from the streptomycin library mediated three hydrogen interactions, initially at 10 ns. However, after 15 ns of the simulation, five hydrogen bonds were observed constantly. ZINC08219868 (Phytoene) and ZINC08214943 (Lycopene) formed no hydrogen bonds, indicating hydrophobic interaction with the LpxC protein (as shown in Figure 8). 

Eventually, the LpxC protein in combination with the nine compounds was found to be stable during the 50 ns simulation. This confirms that the docking interactions that the complexes formed were stable. 

## 3. Discussion

*Campylobacter* infections are usually caused by the consumption of contaminated poultry products [9]. Although the frequency of human sickness caused by oral Campylobacter species is lower than that caused by zoonotic *Campylobacter* species, it is considered that non-*jejuni/coli Campylobacter* illness is underreported due to a lack of good culture-based detection methods [5]. Recently, *C. ureolyticus* has been reported to cause gastroenteritis in both developed and developing countries [3]. 

The current study intends to prioritize potential therapeutic targets, as well as lead drug candidates, against *C. ureolyticus* based on the results of a comprehensive pan-genome investigation. In this study, we used the pan-genome to detect and characterize the *C. ureolyticus* antimicrobial resistome as well. The pan-genome concept has previously been utilized to distinguish between commensal and pathogenic strains [52,53] and we were able to investigate the pan-resistome of the 13 strains as well. The pan-resistome analysis is useful for determining the ARG diversity of the species and revealing the occurrence of species-specific ARGs. Previously, Costa et al. [12] identified that some ARGs were conserved between *C. jejuni* and *C. lari*. In our analysis, we also found conservation of fluoroquinolone-resistant *gyrA* gene between *C. jejuni* and *C. ureolyticus* and its specific mutation T86I. TetM also exists in *C. jejuni* [54], but OXA-85 was unique to *C. ureolyticus* and has only been previously reported in *Fusobacterium nucleatum* subsp. *polymorphum* [21]. 

For drug target mining, the core genome of *C. ureolyticus* was subjected to subtractive genome analysis and 35 targets were obtained. Previously, 34 targets have been reported in *C. jejuni* [55]. Proteins with a role in metabolic pathways (especially lipopolysaccharide synthesis), secondary metabolite synthesis, two-component systems, and multi-drug resistance were identified as drug targets and some of them were similar to previously reported targets in *C. jejuni*. Eventually, UDP-3-O-acyl-N-acetylglucosamine deacetylase (LpxC) was selected as a potent drug target against *C. ureolyticus*. This protein previously has been identified as a potential drug target against *C. jejuni* [55], *Pseudomonas aeruginosa* [32], and several Gram-negative pathogens [56]. The 3D structure of this protein was modeled and a library of LpxC zinc inhibitors, probiotics, and streptomycins (957 total compounds) was screened against it. ADMET profiling was applied to identify associated adversities. This helps to identify decisive factors in relation to a molecule’s potential to be further processed for use as a drug. Harmonizing toxicity and ADME helps to summarize the criteria for the ideal compound, and thus compounds can be narrowed down at the initial screening stage to decide whether to proceed further. Our studied compounds showed several good properties, and their scaffolds could be utilized further for designing and optimizing drugs against *C. ureolyticus* and possibly other bacteria harboring LpxC.

Despite the powerful potential of the screening and validation of binding properties with rigorous simulations, our study has its limitations. Computational predictions do not give 100% accurate results and failure is possible to some extent. Therefore, the results need to be interpreted with caution for use in clinical settings and pharmaceutical endeavors. Lab validations must be carried out before moving on to clinical trials of high-throughput screened molecules. Better algorithms and more computational power may be available in the future to resolve unstudied and necessary features of drug-like molecules but current investigations are rife with errors. This should not, however, discourage studies at the computational level; rather, it should stimulate further endeavors.

## 4. Materials and Methods

### 4.1. Data Retrieval 

The complete genomes of *C. ureolyticus* strains (*n* = 22) were retrieved from the NCBI database [57]. Redundant species data were removed and 13 genomes were left (details in Appendix A). 

### 4.2. Pan-Genomic Analysis 

The pan-genomic analysis was performed using the BPGA tool [58], the parameters for which were defined in our previous work [59]. The clustering of inputted FASTA files was performed using the USERACH algorithm [60], with defined cutoff values of 70% for homologous genes. The pan- and core genome dot plots were generated. Furthermore, the alignment of core, accessory, and unique genomes was performed using the MUSCLE tool [61] with default parameters. UPGMA based on maximum likelihood was used to infer the phylogenetic relationships between strains. Additionally, the resistome of the identified core, unique, and accessory genes was investigated via the Comprehensive Antibiotic Resistance Database (CARD) [62]. CARD employs the automated BLASTp algorithm and a cutoff of 70% for alignment. The homologous gene set was functionally annotated through the Clusters of Orthologous Groups of proteins (COG) database [63]. The genome fractions were annotated for gene functions. 

### 4.3. Drug Target Prioritization

The obtained core genome was subjected to the standalone CD-HIT program [64] and homologous gene products with 60% similarity exclusion criteria were clustered. CD-HIT also reduces redundancy. Using an in-house script, all of the clusters were retrieved from the CD-HIT output file and further investigated for gene essentiality. The obtained data were BLAST-searched against the CEG [65] and DEG [66] databases (with an E-value of 10^−10^) and a bit score of 100. The proteins found essential in both databases were used for further target identification. The significantly preferable drug targets must be non-homologous to the human genome in order to avoid the cross-reactivity of drug candidates. For this purpose, the translated products of the obtained sets of genes were subjected to BLASTp against the human proteome (conditions: E-value: 10^−3^; gap penalty: 11; gap extension penalty: 1). 

Furthermore, a comparative analysis was performed between obtained non-homologous proteins and the proteomes of the 83 useful species of human microbial gut flora. The acquired set of proteins was processed against the DrugBank database. The identified druggable targets were also filtered through the virulence factor database (VFDB) [67] using an E-value of 10^−3^.

### 4.4. Structural Modeling 

The structure of the selected protein LpxC was modeled using SWISS-MODEL [68]. This tool uses a homology-based approach for deciphering protein structures. The final model was selected based on its percent identity and query coverage to its template LpxC protein from *Escherichia coli* (PDB ID: 4MDT). Furthermore, the validation of the modeled structure was achieved using a Ramachandran plot. Secondary structure evaluation, Z-score prediction, and stereochemical quality (via PROCHECK) were also assessed through the SWISS-MODEL server [69]. The secondary structure and the disordered regions were predicted using the Phyre2 server [70].

### 4.5. Virtual Screening

The interactions between the screened streptomycins (*n* = 737), postbiotics (*n* = 78), and LpxC zinc database inhibitors (*n* = 142) against the LpxC protein were appraised using a molecular docking-based screening approach [71]. Using MOE v2019 software, the pre-docking protonation of LpxC was carried out with the following parameters: flip: all atoms; atoms: all atoms; titrate: all atoms; solvent: 80, disconnected metal treatment: enabled; temperature = 300K; salt = 0.1; pH = 7; van der Waals: 800R3, with cutoff (A): 10; dielectric: 1, 1; protection = none; electrostatics: GB/VI, with cutoff (A): 15. Correspondingly, the following parameters were set for energy minimization: forcefield: Amber99; gradient: 0.05; fix hydrogen and partial charges = yes. The 957 compounds from the mentioned libraries were taken and screened against LpxC. The screening parameters were: placement = Triangle Matcher; refinement = forcefield; rescoring 1 = London dG; rescoring 2 = affinity dG. The top docked compounds were chosen based on their S-values. The interactions of the docked protein–ligand complexes were examined to explore the hydrogen bonding and hydrophobic interactions among receptors and ligand atoms within a range of 5 Å [72]. 

### 4.6. ADMET Profiling of the Shortlisted Drug Candidates

After the molecular docking study, pharmacokinetic properties of the shortlisted lead drug compounds, such as absorption, distribution, metabolism, excretion (ADME), and toxicology, were determined using Swiss ADME [73]. Following this, the selected compounds were processed using the pkCSM tool (http://biosig.unimelb.edu.au/pkcsm/ (accessed on 28 December 2021)) to determine the optimal drug candidate with high penetration and the least number of side effects. SwissADME presented skin permeation values, and pkCSM provided drug tolerance values for a range of organisms, including humans. Drug safety evaluation is critical for new drugs. Early knowledge of medication toxicity and side effects is crucial for drug induction in the development pipeline [74,75]. With this in mind, the highest tolerated dosages, the impacts on various species, and the excretion of drug characteristics were all determined.

### 4.7. Molecular Dynamics (MD) Simulation

The stability and flexibility analyses of the identified compounds against the shortlisted drug targets were carried out through MD simulation studies. MM/PBSA values were calculated using MOE software [76], while the atom-scale MD simulations were carried out with the GROMACS 5.1.2 package24 using the gromos54a71 forcefield. The ligand topology parameters were generated using the Automated Topology Builder (ATB v3.0) server. Using periodic boundary conditions, the complex was placed at a distance of 1.0 from the box edge in a dodecahedron box, with the addition of solvent molecules from the SPC water model. The insertion of counter ions into the solvated system helps to neutralize the entire system. The neutralized system was minimized using the steepest descent method, with a maximum force of 1000 kJ mol^−1^. The minimized systems were passed for 100 ps of NVT and NPT equilibration before the production dynamics to raise the system temperature by 300K and maintain a constant pressure of 1 bar for the system utilizing thermostat and Berendsen barostat algorithms. The Linear Constraint Solver (LINCS) algorithm was used to constrain all the bonds, and the Particle Mesh Ewald (PME) method was employed to compute the long-range electrostatics with a cutoff value of 1.0 nm. Finally, a 50 ns production run was carried out, with coordinates and energies being saved every 10 ps in the output trajectory file, as per the procedure. Root Mean Square Deviation (RMSD), Root Mean Square Fluctuation (RMSF), and Radius of Gyration (Rg) were plotted to evaluate the system’s stability.

## 5. Conclusions

To the best of our knowledge, the current study is the first to map the resistome of *C. ureolyticus* as well as drug targets for this species. Resistance profiling is not only useful for the scientific community working in this domain but also for clinicians and health policymakers. Our computational analysis will aid the scientific community in further investigating the proposed therapeutic drug targets and inhibitors using experimental methodologies. The identified therapeutic proteins might be studied further in experimental investigations to prevent *C. ureolyticus*-mediated gastroenteritis, along with the compounds that we screened against the LpxC enzyme. The prioritized compounds ZINC26844580, ZINC13474902, ZINC13474878, Notoginsenoside St-4, Asiaticoside F, Paraherquamide E, Phytoene, Lycopene, and Sparsomycin need to be tested further in vivo and in vitro for their druggability. 

## Figures and Tables

**Figure 1 antibiotics-11-00680-f001:**
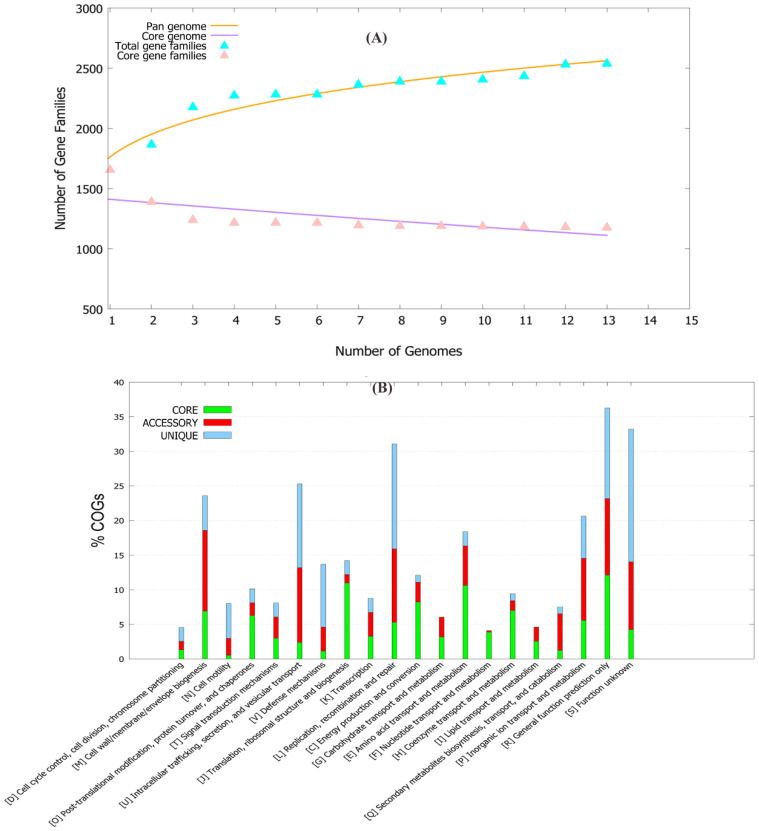
(**A).** A dot plot of the pan-genome vs. the core genome of 13 *C. ureolyticus* strains. (**B**) COG distribution of genome fractions for *C. ureolyticus* strains.

**Figure 2 antibiotics-11-00680-f002:**
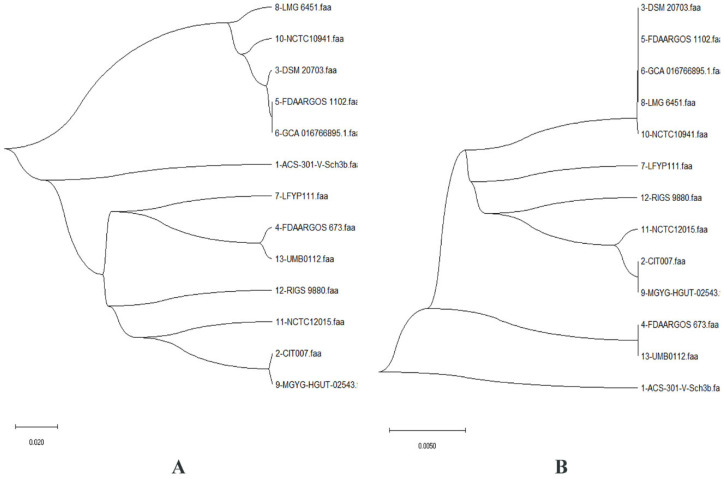
Curved phylogenomic trees based on: (**A**) the pan-genome of 13 *C. ureolyticus* strains (SBL is 0.71, while the distance in millions of years is 0.02); and (**B**) the core genome of the studied strains (SBL value is 0.08, while the distance in millions of years is 0.005).

**Figure 3 antibiotics-11-00680-f003:**
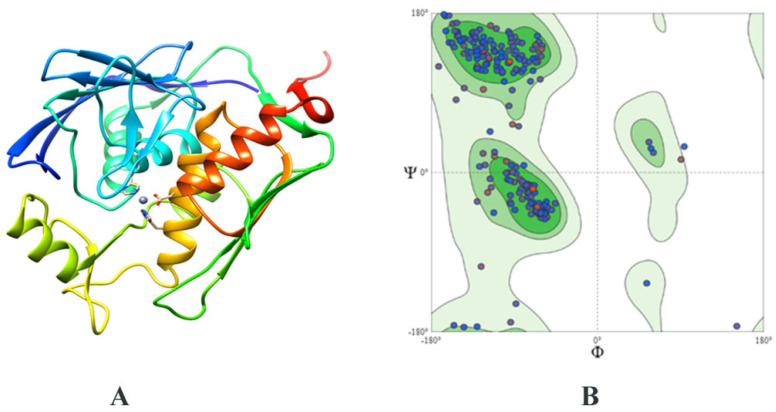
(**A**) Modeled structure of the LpxC protein. (**B**) Ramachandran plot using PROCHECK, showing 93% residues in the favored region.

**Figure 4 antibiotics-11-00680-f004:**
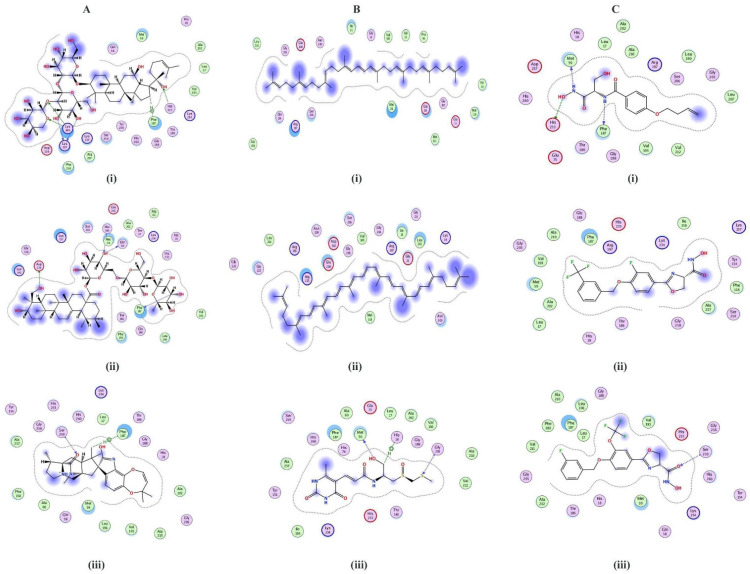
Two-dimensional depictions of shortlisted compounds from: (**A**) the postbiotics library, i.e., (**i**) Notoginsenoside St-4, (**ii**) Asiaticoside F, and (**iii**) Paraherquamide E; (**B**) the streptomycin library, i.e., (**i**) ZINC08219868 (Phytoene), (**ii**) ZINC08214943 (Lycopene), and (**iii**) ZINC04742519 (Sparsomycin); and (**C**) the ZINC LpxC library, i.e., (**i**) ZINC26844580, (**ii**) ZINC13474902, and (**iii**) ZINC13474878.

**Figure 5 antibiotics-11-00680-f005:**
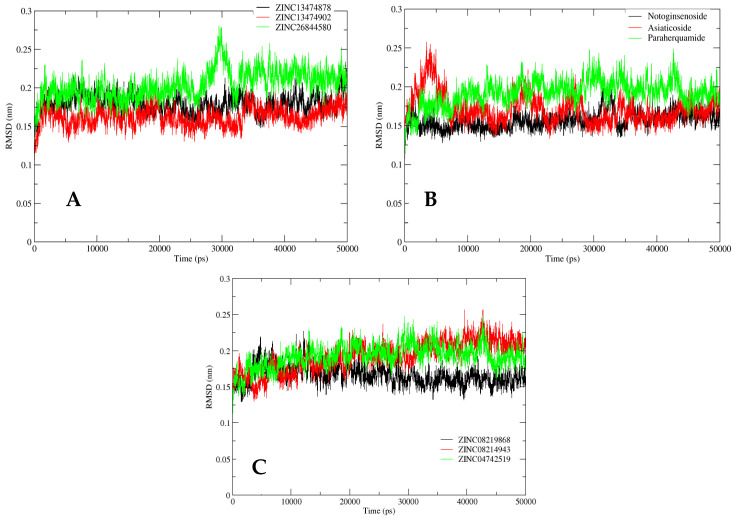
RMSD analysis for the shortlisted compounds: (**A**) ZINC26844580 (green), ZINC13474902 (red), and ZINC13474878 (black); (**B**) Notoginsenoside St-4 (black), Asiaticoside F (red), and Paraherquamide E (green); and (**C**) ZINC08219868 (Phytoene) (black), ZINC08214943 (Lycopene) (red), and ZINC04742519 (Sparsomycin) (green).

**Figure 6 antibiotics-11-00680-f006:**
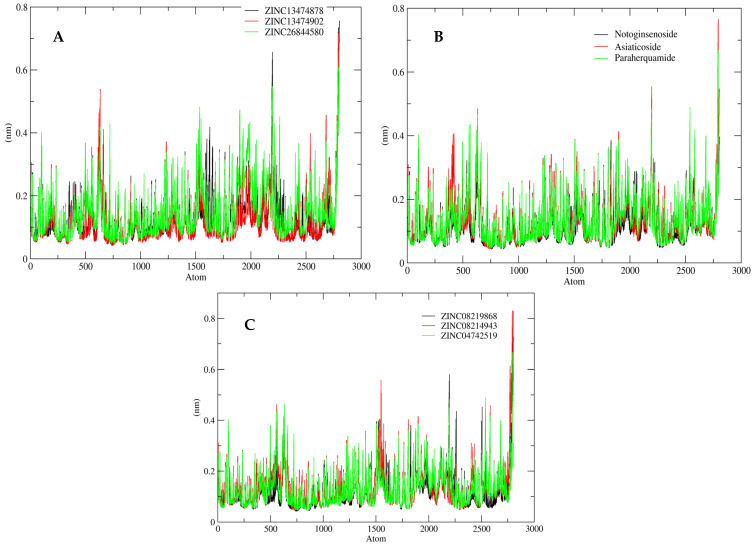
RMSF analysis for the shortlisted compounds: (**A**) ZINC26844580 (green), ZINC13474902 (red), and ZINC13474878 (black); (**B**) Notoginsenoside St-4 (black), Asiaticoside F (red), and Paraherquamide E (green); and (**C**) ZINC08219868 (Phytoene) (black), ZINC08214943 (Lycopene) (red), and ZINC04742519 (Sparsomycin) (green).

**Figure 7 antibiotics-11-00680-f007:**
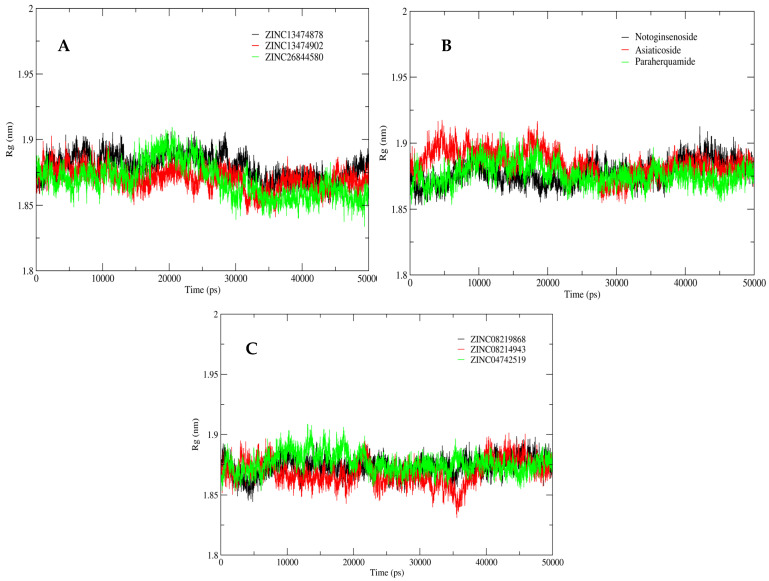
Rg analysis for the shortlisted compounds: (**A**) ZINC26844580 (green), ZINC13474902 (red), and ZINC13474878 (black); (**B**) Notoginsenoside St-4 (black), Asiaticoside F (red), and Paraherquamide E (green); and (**C**) ZINC08219868 (Phytoene) (black), ZINC08214943 (Lycopene) (red), and ZINC04742519 (Sparsomycin) (green).

**Figure 8 antibiotics-11-00680-f008:**
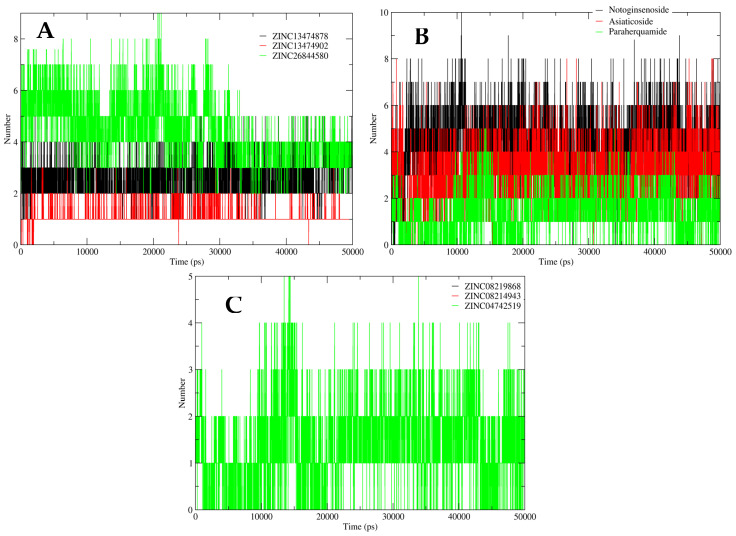
Hydrogen bond analysis for the shortlisted compounds: (**A**) ZINC26844580 (green), ZINC13474902 (red), and ZINC13474878 (black); (**B**) Notoginsenoside St-4 (black), Asiaticoside F (red), and Paraherquamide E (green); and (**C**) ZINC08219868 (Phytoene) (black), ZINC08214943 (Lycopene) (red), and ZINC04742519 (Sparsomycin) (green).

**Table 1 antibiotics-11-00680-t001:** Drug targets of *C. urolyticus* that are virulent in nature. The selected target is shown in bold.

Serial No.	Protein	Functional Category	Accession	No. of Amino Acids
1	Multidrug efflux RND transporter permease subunit	Signaling and cellular processes	WP_016646469.1	1053
2	Formate dehydrogenase subunit alpha	Carbohydrate metabolism	WP_081617940.1	974
3	Formate dehydrogenase subunit alpha	Energy metabolism	WP_081617935.1	935
4	Thiosulfate reductase PhsA		WP_024962542.1	761
5	Urease subunit alpha	Nucleotide metabolism	WP_050333258.1	571
6	TolC family protein	Cellular process	WP_018713635.1	481
7	Murein biosynthesis integral membrane Protein MurJ	Metabolism	WP_101637374.1	470
8	3-deoxy-7-phosphoheptulonate synthase class II		WP_018713201.1	447
9	Glutamyl-tRNA reductase		WP_101637465.1	436
10	Type II secretion system F family protein	Environmental information processing	WP_101636691.1	415
11	Lipid IV(A) 3-deoxy-D-manno-octulosonic acid transferase	Metabolism	WP_018712363.1	382
12	Efflux RND transporter periplasmic adaptor subunit	Genetic information processing	WP_101637141.1	371
13	Lipid-A-disaccharide synthase	Metabolism	WP_018713606.1	347
14	KpsF/GutQ family sugar-phosphate isomerase		WP_018713574.1	320
15	UDP-3-O-(3-hydroxymyristoyl)glucosamine N-acyltransferase		WP_101637361.1	317
16	UDP-3-O-acyl-N-acetylglucosamine deacetylase (LpxC)		WP_050333632.1	296
17	c-type cytochrome		WP_101637480.1	287
18	Pantoate--beta-alanine ligase		WP_016646546.1	273
19	3-deoxy-manno-octulosonate cytidylyltransferase		WP_018713833.1	240
20	Carbonic anhydrase		WP_018713612.1	227
21	MotA/TolQ/ExbB proton channel family protein	Signaling and cellular processes	WP_018713282.1	191
22	YceI family protein	-	WP_024962547.1	188
23	Biopolymer transporter ExbD	Signaling and cellular processes	WP_018713283.1	130
24	Aspartate 1-decarboxylase	Metabolism	WP_018713496.1	115
25	Urease subunit beta	Nucleotide metabolism	WP_018713462.1	104
26	Urease subunit gamma	Nucleotide metabolism	WP_018713461.1	100

**Table 2 antibiotics-11-00680-t002:** Molecular docking analysis of shortlisted compounds from the studied libraries.

S. No.	Compounds	Ligand	Receptor	Interaction	Distance	E (kcal/mol)	S-Score
LpxC ZINC database inhibitors							
1	ZINC26844580	N12	O PHE187	H-donor	3.43	−0.8	−7.42
		N18	O MET59	H-donor	2.95	−2.3	
		O19	NE2 HIS233	H-donor	2.92	−3.1	
2	ZINC13474902	Hydrophobic interactions					−7.05
3	ZINC13474878	O1	CA SER259	H-acceptor	3.44	−0.8	−6.90
Postbiotics							
1	Notoginsenoside St-4	O63	O LYS157	H-donor	2.89	−2.6	−8.59
		O64	O LYS157	H-donor	3.12	−1.0	
		O35	NE2 HIS233	H-acceptor	2.93	−2.3	
		O53	NZ LYS140	H-acceptor	3.12	−0.5	
		O55	NZ LYS140	H-acceptor	3.00	−5.7	
		C26	6-ring PHE187	H-pi	4.32	−0.6	
2	Asiaticoside F	O79	OD1 ASP139	H-donor	2.89	−2.0	−8.43
		O94	O GLN58	H-donor	3.26	−0.5	
		C112	O PHE187	H-donor	3.30	−0.5	
		O94	N HIS260	H-acceptor	3.40	−0.7	
3	Paraherquamide E	O43	CA SER259	H-acceptor	3.43	−0.6	−8.02
		C18	6-ring PHE 187	H-pi	3.81	−0.7	
Streptomycin compounds							
1	ZINC08219868 (Phytoene)	Hydrophobic interactions					−7.20
2	ZINC08214943 (Lycopene)	Hydrophobic interactions					−7.03
3	ZINC04742519 (Sparsomycin)	O27	O MET59	H-donor	2.84	−0.7	−7.01
		S37	CA GLY205	H-acceptor	4.06	−0.7	
		C24	5-ring HIS18	H-pi	4.82	−0.5	

**Table 3 antibiotics-11-00680-t003:** ADMET properties analysis of the nine compounds shortlisted.

Name	Water Solubility	CaCo2 Permeability	HIA	Skin Permeability	Max. Tolerated Dose (Human)	Minnow Toxicity	T. Pyriformis Toxicity	Oral Rat Acute Toxicity (LD50)	Hepatotoxic
ZINC26844580	−2.348	−0.049	51.882	−2.898	1.177	2.262	0.269	2.238	Yes
ZINC13474902	−5.016	1.037	91.596	−3.066	0.193	2.117	0.422	2.983	Yes
ZINC13474878	−4.928	1.103	90.066	−2.993	0.272	2.587	0.356	2.96	Yes
Notoginsenoside St−4	−2.938	−1.241	0	−2.735	−1.755	10.516	0.285	2.738	No
Asiaticoside F	−2.922	−1.039	43.096	−2.735	−0.991	11.112	0.285	2.738	No
Paraherquamide E	−4.377	1.003	93.521	−3.226	−0.877	2.209	0.294	3.716	Yes
ZINC08219868 (Phytoene)	−6.345	1.28	91.213	−2.737	−0.33	−5.751	0.288	2.036	No
ZINC08214943 (Lycopene)	−6.514	1.317	93.238	−2.783	−0.447	−5.243	0.291	2.105	No
ZINC04742519 (Sparsomycin)	−2.348	−0.238	44.731	−3.08	1.279	2.726	0.146	2.11	Yes

**Table 4 antibiotics-11-00680-t004:** MM/PBSA values of the selected compounds.

Name	Ligand	Protein	Protein−Ligand Complex
ZINC1347878	0.04	−19.31	−18.95
ZINC13474902	0.03	−19.31	−18.95
ZINC26844580	0.14	−19.31	−19.03
Notoginsenoside St-4	−0.85	−19.31	−19.19
Asiaticoside F	−0.82	−19.31	−19.20
Paraherquamide E	−0.07	−19.31	−18.89
Phytoene	−0.96	−19.31	−19.46
Lycopene	−0.92	−19.31	−19.54
Sparsomycin	0.05	−19.31	−18.97

## Data Availability

Publicly analyzed datasets used in this study can be found in NCBI GenBank by reference to the details provided in Appendix A. Any other raw or generated data are available from the corresponding authors without reservation.

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
