# Peer review of "Identification of Therapeutic Targets in an Emerging Gastrointestinal Pathogen Campylobacter ureolyticus and Possible Intervention through Natural Products"

_antibiotics, 2022, doi:10.3390/antibiotics11050680_

Round 1

Reviewer 1 Report

The MS describes a combination of genomics and docking for antibiotic discovery. The overall MS quality is bad. Some Figs are unreadable (1,2,4); some look like SI (5-8). The number of references (79!) brings to mind a review paper.

The general problem is that no predicted drug candidate was tested. Even known LpxC inhibitors were not tested in C. ureolyticus. Moreover, some of the predicted molecules are potent antibiotics with documented antimicrobial mechanism (translational inhibitor Sparsomycin) were mentioned as predicted LpxC inhibitors. These evidences are very ambiguous and MUST be proven.

Author Response

Reviewer 1

The MS describes a combination of genomics and docking for antibiotic discovery. The overall MS quality is bad. Some Figs are unreadable (1,2,4); some look like SI (5-8). The number of references (79!) brings to mind a review paper.

Authors response: We have now redone the figures and tried to reduce the number of references but it seems that every reference is important and eliminating it would impact the manuscript. We would be thankful if the reviewer kindly let us know which reference can be eliminated and we will remove it.

The general problem is that no predicted drug candidate was tested. Even known LpxC inhibitors were not tested in C. ureolyticus. Moreover, some of the predicted molecules are potent antibiotics with documented antimicrobial mechanism (translational inhibitor Sparsomycin) were mentioned as predicted LpxC inhibitors. These evidences are very ambiguous and MUST be proven.

Authors response: The title of our paper states ‘possible intervention through natural products’ and not proven intervention through natural products’. The antibiotic sparsomycin is a natural product (a metabolite produced by Streptomyces sparsogenes), and we screened the library of such molecules, which were produced by Streptomyces. Docking is a proven technique for inhibitor screening and simulation was used for validation of binding parameters. This is how sparsomycin prioritized and it is novel finding for LpxC inhibition, as it has not previously been reported to inhibit this class of enzymes.

We are eager to test how these compounds behave in the live organism, on cell lines and in mouse models. Such lab analysis constitutes another study and would be done in due course, subject to availability of funds. Till then, prediction based analysis is feasible and we cannot ignore such a possibility of binding based on rules of physics and chemistry.

Reviewer 2 Report

The introduction can be improved considering data and other informations about Campylobacter ureolyticus in Humans and animals.

Author Response

Reviewer 2

The introduction can be improved considering data and other informations about Campylobacter ureolyticus in Humans and animals.

Authors response: We have now added new information. Text is reproduced below:

Farm animals are the primary reservoir for Campylobacter sp. infections and the primary cause of campylobacteriosis. Farm animals are also the leading source of bacterial food poisoning and Campylobacter gastrointestinal illnesses worldwide. Campylobacter foodborne illness is a concern and an expensive burden for the human population, ac-counting for 8.4% of all diarrhoea cases worldwide. In most cases, C. ureolyticus has been recovered from human samples, with just one report of C. ureolyticus isolation from healthy horse endometria. Following that, a retrospective investigation of over 7,000 patients with diarrhea found C. ureolyticus in 23.8 % of Campylobacter-positive samples, marking the first discovery of C. ureolyticus in the faeces of gastroenteritis patients and highlighting the species involvement as an emerging enteric pathogen [3-5].

Reviewer 3 Report

Dear Authors,
the manuscript requires corrections in several places, e.g. in the names of
bacteria.
C. ureolyticus should always be written in italics (e.g. under Figure 1).
The name of the disease caused by the genus Campylobacter is
campylobacteriosis, but written without italics.
The name of the bacterium Fusobacterium nucleatum subspecies polymorphum is
also incorrect, only the word "subspecies" should remain without italics, all
the rest should be written in italics. The Figures in the manuscript are completely illegible, out of focus
(even when enlarged) and require correction.
Gene names (e.g. gyrA) should be in italics. We use the name Streptomyces spp. (incorrectly used name). We never use the
name of Campylobacter or Streptomyces, we always mean species, so we add sp.
or spp.
Dear Authors, the discussion begs to be expanded.
It should be added here whether a similar therapeutic profile was made for
any other bacteria. Has it been tested under laboratory conditions? Has it
brought the expected result? Are such studies known? Are yours pioneering?
In this way, you will show the importance of your simulations and the actual
contribution to the search for new drugs.

Best regards.

Author Response

Reviewer 3

The manuscript requires corrections in several places, e.g. in the names of bacteria. C. ureolyticus should always be written in italics (e.g. under Figure 1).

Authors response: Corrected in revise manuscript

The name of the disease caused by the genus Campylobacter iscampylobacteriosis, but written without italics.

Authors response: Edited all the genus names in italics in revised manuscript.

The name of the bacterium Fusobacterium nucleatum subspecies polymorphum is
also incorrect, only the word "subspecies" should remain without italics, all
the rest should be written in italics.

Authors response: Corrected

The Figures in the manuscript are completely illegible, out of focus (even when enlarged) and require correction. 

Authors response: Corrected. All of them are 300 dpi, but too many compound information or data in some figures makes them bit difficult to see clearly but we do not have any solution to that.

Gene names (e.g. gyrA) should be in italics. 

Authors response: Corrected

We use the name Streptomyces spp. (incorrectly used name). We never use the name of Campylobacter or Streptomyces, we always mean species, so we add sp. or spp.

Authors response: Corrected

Dear Authors, the discussion begs to be expanded. It should be added here whether a similar therapeutic profile was made for any other bacteria. Has it been tested under laboratory conditions? Has it
brought the expected result? Are such studies known? Are yours pioneering? In this way, you will show the importance of your simulations and the actual contribution to the search for new drugs.

Authors response: We have already mentioned in introduction that such in silico studies have been published for various organisms, including several of our own group. However, we could not find any in Campylobacter ureolyticus as this is the first study to implement in silico pipeline for target mining and drug discovery against this bacteria. Ours is pioneering and we have mentioned in conclusion that that ‘To the best of our knowledge, the current study mapped the first resistome of C. ureolyticus as well as drug targets for this species.’ LpxC has been implied as drug target in other species as mentioned on page 15. Text is reproduced below:

‘Many effective LpxC inhibitors have been identified [31, 32], with a range of chemical scaffolds and antibiotic profiles[31] but none of them have attained approval as antibacterial moieties[31]. LpxC inhibitors have not been reported for C. ureolyticus.’

Reviewer 4 Report

In the present study, the authors investigate multiple isolates of an emerging gastrointestinal C. ureolyticus to better understand its pathogenesis and possible intervention through natural products.  Furthermore, this work in present form presents some imperfections according to the following comments:

  • The major remark is the inability to read Figures 1; 2; 5; 7 and 8 which prevents reading the paper well
  • Page 8; figure 3: in the title is noted “Modeled structure of the protein” You must specify which protein you are talking about.
  • In parts of the Materials and Methods section such as Page 8; lines 175-177 the authors discuss the results. Despite they have a separate discussion section that needs further improvement
  • Page 10; line 208: compounds must be attributed with their figure in (B)
  • Page 15; line 340: You must put the corresponding number of the Costa and Iraola (2019) reference, same remark for line 345 reference (Voha, Docquier, Rossolini, & Fosse, 2006).

Author Response

Reviewer 4

In the present study, the authors investigate multiple isolates of an emerging gastrointestinal C. ureolyticus to better understand its pathogenesis and possible intervention through natural products.  Furthermore, this work in present form presents some imperfections according to the following comments:

  • The major remark is the inability to read Figures 1; 2; 5; 7 and 8 which prevents reading the paper well

Authors response: Corrected. Now all figures are 300 dpi resolution.

  • Page 8; figure 3: in the title is noted “Modeled structure of the protein” You must specify which protein you are talking about.

Authors response: Corrected. The structure of the selected protein LpxC was modeled using SWISS-MODEL.

  • In parts of the Materials and Methods section such as Page 8; lines 175-177 the authors discuss the results. Despite they have a separate discussion section that needs further improvement

Authors response: Some things do not make sense if not explained a bit, this is why we briefly discuss them along side results and for the detailed discussion, comparison with previous literature etc, we have a separate section.

  • Page 10; line 208: compounds must be attributed with their figure in (B)

Authors response: Done in legend.

  • Page 15; line 340: You must put the corresponding number of the Costa and Iraola (2019) reference, same remark for line 345 reference (Voha, Docquier, Rossolini, & Fosse, 2006).

Authors response: Done.

Round 2

Reviewer 1 Report

All comments generally remained unaddressed.

Reviewer 3 Report

Thanks for the corrections made. Regards

Reviewer 4 Report

The authors have responded and modified their paper according to the requested recommendations, I recommend the publishing of the paper in this form.